# Identification of suitable reference genes for normalization of reverse transcription quantitative real-time PCR (RT-qPCR) in the fibrotic phase of the bleomycin mouse model of pulmonary fibrosis

Oula Norman, Jarkko Koivunen, Joni M. Mäki, Taina Pihlajaniemi, Anne Heikkinen ⓘ *

ECM-Hypoxia Research Unit, Faculty of Biochemistry and Molecular Medicine, University of Oulu, Oulu, Finland

* anne.heikkinen@oulu.fi

## Abstract

Idiopathic pulmonary fibrosis (IPF) is a severe lung disease with a poor prognosis and few treatment options. In the most widely used experimental model for this disease, bleomycin is administered into the lungs of mice, causing a reaction of inflammation and consequent fibrosis that resembles the progression of human IPF. The inflammation and fibrosis together induce changes in gene expression that can be analyzed with reverse transcription quantitative real-time PCR (RT-qPCR), in which accurate normalization with a set of stably expressed reference genes is critical for obtaining reliable results. This work compares ten commonly used candidate reference genes in the late, fibrotic phase of bleomycin-induced pulmonary fibrosis and ranks them from the most to the least stable using NormFinder and geNorm. *Sdha*, *Polr2a* and *Hprt* were identified as the best performing and least variable reference genes when alternating between normal and fibrotic conditions. In order to validate the findings, we investigated the expression of *Tnf* and *Col1a1*, representing the hallmarks of inflammation and fibrotic changes, respectively. With the best three genes as references, both were found to be upregulated relative to untreated controls, unlike the situation when analyzed solely with *Gapdh*, a commonly used reference gene. We therefore recommend *Sdha*, *Polr2a* and *Hprt* as reference genes for RT-qPCR in the 4-week bleomycin challenge that represents the late fibrotic phase.

## Introduction

Idiopathic pulmonary fibrosis (IPF) is the most common of the interstitial pneumonias and is a progressive incurable disease with a poor prognosis. Its yearly incidence in Europe and North America is estimated to be 2.8–18 cases per 100 000 persons, which is similar to diseases such as stomach or brain cancer [1]. The cause of the disease has been viewed as being a combination of various genetic and environmental cues that lead to the proliferation of fibroblasts

**Data Availability Statement:** All relevant data are within the manuscript and its Supporting Information files.

**Funding:** This study was supported by grants from the Sigrid Jusélius Foundation (TP) (https://www.sigridjuselius.fi/en/), the Jane and Aatos Erkko Foundation (TP) (https://jaes.fi/en/) and the Emil Aaltonen Foundation (ON) (https://emilaaltonen.fi/). The funders had no role in study design, data collection and analysis, decision to publish, or preparation of the manuscript.

**Competing interests:** The authors have declared that no competing interests exist.

and the accumulation of pathological extracellular matrix, a process in which changes in gene expression are instrumental, ultimately leading to the alveolar space being replaced with connective tissue. Multiple attempts have been, and continue to be, made to develop an anti-fibrotic drug, but despite pre-clinical studies showing positive effects, most compounds have not shown comparable performance in clinical trials [2] and to date only two anti-fibrotic pharmaceuticals that retard the progression of IPF have been approved for use [3]. Thus we still need a better understanding of the disease and more effective therapy, as median survival remains at the level of 2–4 years after diagnosis [1], thereby empowering further investigations.

Reverse transcription quantitative real-time PCR (RT-qPCR) is a widely used method for assessing gene expression, the key to its accuracy being valid normalization of the expression of the gene of interest to a set of stably expressed reference genes in order to minimize the effects of differences in the quantity and quality of the RNA between individual samples. The reference genes are often housekeeping genes that have a role in the cell's basic vital activities such as energy metabolism or protein synthesis. A systematic review by Chapman and Waldenström [4] shows that the most commonly used reference genes are beta-actin (*Actb*), glyceraldehyde-3-phosphate dehydrogenase (*Gapdh*) and the ribosomal RNA 18S subunit (*Rn18s*), although even their expression does not always remain stable but often varies between experimental groups. Thus the optimal reference genes have to be verified for the experiment in question in order to achieve accurate results. *Gapdh*, in particular, has numerous pseudogenes that may confound the analysis [5, 6]. In addition, it is upregulated under hypoxic conditions and most inflammatory tissues exhibit a degree of hypoxia due to edema [7]. In the case of *Actb* attention should be paid to primer design, as it shares 91% of its sequence with leucine-rich repeat-containing 58 (*Lrrc58*) [6].

Different methods have been developed to determine which genes are most suitable for normalization, some of the most commonly used algorithms being NormFinder [8] and geNorm [9]. NormFinder employs a model-based approach that estimates intra- and intergroup variance for each candidate gene. This approach relies on the assumption that the average expression of the candidate genes will be the same in both experimental groups. geNorm uses a pairwise comparison approach in which the candidate genes are ranked according to the similarity of their expression profiles. This method requires that the candidate genes be regulated independently of each other. Unlike NormFinder, geNorm does not compare the expression levels of the candidate genes between experimental groups but instead comparisons are made between all the samples. RNA sequencing has also been used for finding stably expressed genes, but Sampathkumar et al. [10] recently demonstrated that the results are similar to the conventional approach.

Bleomycin is an anti-neoplastic drug that has a potential to cause lung fibrosis when used in large doses [11]. The histological changes it causes in mouse lungs are relatively similar to those seen in human IPF and thus the bleomycin model is the most widely used experimental animal model for studying pulmonary fibrosis [12]. The reaction to the administration of bleomycin to the lungs consists of an initial phase of acute lung injury, inflammation and the development of fibrosis that lasts for around two weeks, followed by a phase of established fibrosis from two to four weeks after bleomycin administration and ultimately slow resolution of the fibrosis [2]. A previous study has explored suitable reference genes in the initial inflammatory phase of the bleomycin mouse model [13], but to the best of our knowledge there are no reports on reference genes suitable for the normalization of RT-qPCR in the fibrotic phase of the bleomycin model by comparison with the untreated state. We present here data on the expression of ten commonly used reference genes during the fibrotic phase of the bleomycin mouse model and rank them for their utility in normalizing gene expression.

## Materials and methods

### Ethics approval

Permission for the administration of bleomycin and the induction of fibrosis was obtained from the Finnish Animal Care and Use Committee (ESAVI/8179/04.10.07/2017). All the mouse experiments complied with the European Community Council Directive on the protection of animals used for scientific purposes (September 22, 2010; 2010/63/EEC), national legislation and the regulations for the care and use of laboratory animals.

### The bleomycin mouse model of pulmonary fibrosis

The mice used here were part of a larger study on the effect of collagen XIII deficiency on pulmonary fibrosis and were wild-type littermates of collagen XIII (*Col13a1*) knockout mice of both sexes. The generation of the mice is described in detail by Latvanlehto et al. [14]. The genetic background of the mice originates from sv129 embryonic stem cells followed by cross breeding with the C57BL/6JOlaHsd strain for 17 generations and the C57BL/6NClr strain for 3–4 generations. The mice were maintained in a specific pathogen free (SPF) facility at +21˚C with a 12-hour light-dark cycle and were fed Teklad global 18% protein rodent diet with *ad libitum* access to food and water. From 6 weeks of age onwards the mice were transferred to animal research facilities and the diet was supplemented with banana and hazelnut cocoa spread to minimize malnutrition and mortality during the development of pulmonary fibrosis. 8-week-old mice were anesthetized with isoflurane and given 1,25 U/kg bleomycin (Baxter) diluted with sterile isotonic saline to a volume of 1.67 microliters/g of body weight by oropharyngeal aspiration as described by Barbayianni et al. [15]. Untreated mice served as controls. After 4 weeks of follow-up the mice were sacrificed by injection of fentanyl, midazolam and medetomidine and exsanguination. The left lung was fixed with intratracheal fixation with 4% paraformaldehyde (PFA) at a pressure of 25 cm $H_2O$ for 2 minutes to open the airways with the right bronchi ligated to prevent fixation. The right lung lobes were dissected, snap frozen in liquid nitrogen and stored at -80˚C. The right postcaval lobe was used for extraction of RNA. After dissection of the right lobes, the left lobe was immersed in 4% PFA for 24 h, embedded in paraffin and used for histological analysis.

### Total RNA extraction and cDNA synthesis

Total RNA was isolated from the frozen right postcaval lung lobes using an RNeasy kit (Qiagen) according to the manufacturer's instructions. All the samples were processed in one batch. The quantity and purity of the RNA were assessed by measuring absorbances at 260 and 280 nm with a Nanodrop spectrophotometer (Thermo Scientific). The quality of the RNA was assessed by capillary electrophoresis using a Bioanalyzer 2100 (Agilent). Based on their fibrotic histological appearance and adequate RNA quality, 9 (4 male, 5 female) bleomycin-treated samples and 10 (5 male, 5 female) untreated samples were selected for use in the analyses. cDNA was synthetized with an iScript cDNA Synthesis kit (Bio-Rad) according to the instructions, using 1000 ng of total RNA and including all the samples in one batch. Genomic DNA was not digested in order to assess whether it affected the results.

### Quantitative real-time PCR

The RT-qPCR reactions were performed using 5 μl iTaq Universal SYBR Green Supermix (Bio-Rad), 1.2 μl of forward and reverse primers at 2.5 μM, 2 μl of cDNA template diluted 1:10 and 0.6 μl of water to a final volume of 10 μl. All the reactions were performed in triplicate and those for a given target gene were all performed on the same plate to minimize batch effects.

Efficiency analyses were mostly performed separately from the sample analyses. The instrument used was a CFX 96 Real-Time PCR system (Bio-Rad). The thermocycling protocol consisted of 3 minutes at 95˚C followed by 40 cycles of denaturation for 15 seconds at 95˚C and annealing for 45 seconds. The annealing temperature was 55˚C for the primers for the succinate dehydrogenase complex subunit A (*Sdha*), 64˚C for the RNA polymerase II subunit A (*Polr2a*) and 60˚C for all the other targets. Melt curves were created by heating from 55˚C to 95˚C with 5-second increments of 0.5˚C and used to assess the specificity of the PCR.

## Primers

The primer sequences were mostly collected from previous publications (Table 1). To minimize unspecific amplification, the primers should be specific to the gene of interest, not form dimers and the amplicon should stretch across an exon-exon junction to distinguish possible genomic DNA amplification. The primer specificity was verified with Primer-BLAST [16]. The primers for *Gapdh* and peptidylprolyl isomerase A (*Ppia*) both amplified several pseudogenes according to BLAST, whereas the other primers were specific to the gene of interest. Amplification of a series of four consecutive 1:5 dilutions of cDNA pooled from the samples with three replicates for each dilution was used to generate a standard curve with quantification cycle (Cq) values on the Y-axis and $\text{Log}_{10}$ of the dilution on the X-axis, a line fitted to the points and primer efficiency (E) calculated with the equation: $E = 10^{-1/slope\ of\ standard\ curve}$ [17]. The standard curves for all the primers, were within the linear dynamic range as indicated by $R^2$ of the standard curves, and the Cq values of all the samples fell inside the linear dynamic range.

## Histological analysis

After fixation the left lungs were embedded in paraffin in the frontal plane and cut into 5 μm sections at 200 μm intervals. The sections were stained with Masson's trichrome. One section immediately before the main bronchus and one after were used for histological analysis. The slides were scanned at 40 x magnification with a NanoZoomer S60 slide scanner (Hamamatsu). Regions of interest with only lung tissue in them were drawn with QuPath [25] and the images downsampled by a factor of 5 with a script [26] to enable processing with ImageJ. Ten 0.9 x 0.9 mm fields were randomly sampled from each section using an ImageJ script modified from [27] and graded using the modified Ashcroft scale [28] independently by 2 observers blinded to the treatment. The scores from all the observers for a given field were averaged, after which the scores for all the fields of the sample were averaged to obtain the score for the entire sample.

## Data analysis

Bio-Rad CFX Maestro 2.2 software was used to analyze the raw data. Automated functions within the software were used for baseline correction, thresholding and Cq determination. For Fig 3, the raw Cq values were corrected for primer-specific PCR efficiency using the formula $Cq_{corrected} = Cq_{raw}\left(\frac{logE}{log2}\right)$. The melt curves were inspected to confirm the specificity of the amplification. NormFinder for R (https://moma.dk/normfinder-software) [8] and GeNorm [9] for Bio-Rad CFX Maestro were used according to their respective instructions. For the NormFinder analysis the raw Cq values were transformed to relative quantities on a linear scale using the formula *Relative quantity (Rq)* $= E^{\Delta Cq}$, where $\Delta Cq = Cq_{sample\ 1} - Cq_{sample\ of\ interest}$. Expression of the collagen I alpha 1 chain (*Col1a1*) and tumor necrosis factor (*Tnf*) normalized to the combination of *Sdha*, *Polr2a* and hypoxanthine guanine phosphoribosyl transferase (*Hprt1*)

**Table 1.** Characteristics of the primers used here.

| Gene symbol | Biological function | Accession number | Amplicon length bp | Forward primer sequence (5'-3') | Forward primer location | Reverse primer sequence (5'-3') | Reverse primer location | Efficiency % | $R^2$ of standard curve | Reference for primer sequence |
|---|---|---|---|---|---|---|---|---|---|---|
| *Actb* | Cytoskeleton | NM_007393.3 | 110 | ACACCCGCCACCAGTTC | Exon 1 | TACAGCCCGGGAGCAT | Exon 2 | 92.3 | 0.999 | [6] |
| *B2m* | Antigen presentation | NM_009735.3 | 200 | TGCTACTCGGCGCTTCAGTC | Exon 1 | AGGCGGGTGAACTGTGTTAC | Exon 2 | 89.5 | 0.997 | [18] |
| *Col1a1* | Connective tissue component | NM_007742.4 | 159 | GGGGCAAGACAGTCATCGAA | Exon 51 | GAGGGAACCAGATTGGGGTG | Exon 51 | 93.1 | 0.999 | [19] |
| *Gapdh* | Glycolysis | NM_001289726.1 | 104 | CATGGCCTTCCGTGTTCCTA | Exon 7 | CCTGCTTCACCACCTTCTTGAT | Exon 7 | 91.2 | 0.999 | [20] |
| *Gusb* | Lysosome component | NM_010368.2 | 86 | CACGGCGATGGACCCAAGAT | Exon 2 | CCCATTCACCCACACAACTGC | Exon 2–3 | 87.4 | 0.990 | [21] |
| *Hprt* | Purine metabolism | NM_013556.2 | 125 | CCCAGCGTCGTGATTAGTGATG | Exon 1–2 | TTCAGTCCTGTCCATAATCAGTC | Exon 2–3 | 92.9 | 0.993 | [13] |
| *Polr2a* | RNA polymerase | NM_001291068.1 | 144 | ATCAACAATCAGCTGCGGCG | Exon 6 | GCCAGACTTCTGCATGGCAC | Exon 7 | 91.1 | 0.995 | [21] |
| *Ppia* | Protein folding | NM_008907.2 | 125 | GAGCTGTTTGCAGACAAAGTTC | Exon 2 | CCCTGGCACATGAATCCTGG | Exon 3 | 96.0 | 0.999 | [22] |
| *Rn18s* | Ribosome component | NR_003278.3 | 123 | GCAATTATTCCCCATGAACG | N/A | GGCCTCACTAAACCATCCAA | N/A | 94.4 | 0.998 | [23] |
| *Sdha* | Citric acid cycle | NM_023281.1 | 103 | CTCTTTTGGACCTTGTCGTCTTT | Exon 9 | TCTCCAGCATTGCCTTAATCGG | Exon 10 | 89.9 | 0.997 | [13] |
| *Tbp* | Transcription factor | NM_013684.3 | 123 | CACCGTGAATCTTGGCTGTAAAC | Exon 4 | CGCAGTTGTTCGTGGCTCTC | Exon 5 | 86.5 | 0.993 | [13] |
| *Tnf* | Inflammation | NM_013693.3 | 117 | TGCCTATGTCTCAGCCTCTT | Exon 1 | GAGGGCCATTTGGGAACTTCT | Exon 2 | 87.2 | 0.997 | [24] |

was calculated using CFX Maestro with the formula:

$$Normalized\ expression_{sample(GOI)} = \frac{Rq_{sample(GOI)}}{\sqrt[3]{Rq_{sample(Sdha)} * Rq_{sample(Polr2a)} * Rq_{sample(Hprt)}}}$$, where GOI = gene of interest,

as proposed by Pfaffl [29] and Vandesompele et al. [9], and expression of *Col1a1* and *Tnf* normalized to *Gapdh* was calculated using the formula

$$Normalized\ expression_{sample(GOI)} = \frac{Rq_{sample(GOI)}}{Rq_{sample(Gapdh)}}.$$

## Statistical analyses and software

The Mann-Whitney U test was used to compare distributions between groups. Data were analyzed with GraphPad Prism (GraphPad Software). GraphPad Prism, CorelDRAW 2021 (Corel) and Corel PHOTO-PAINT 2021 (Corel) were used to create the figures.

## Results

### Level of pulmonary fibrosis

To confirm the suitability of the samples for investigating pulmonary fibrosis, the degree of fibrosis in the lungs was graded with a modified Ashcroft score [28]. The bleomycin-treated samples showed clear fibrotic changes with a mean score of 3.2 (Standard deviation (SD) 0.98), while the untreated mice had a score of 0.4 (SD 0.086) and exhibited no signs of pulmonary fibrosis (Fig 1).

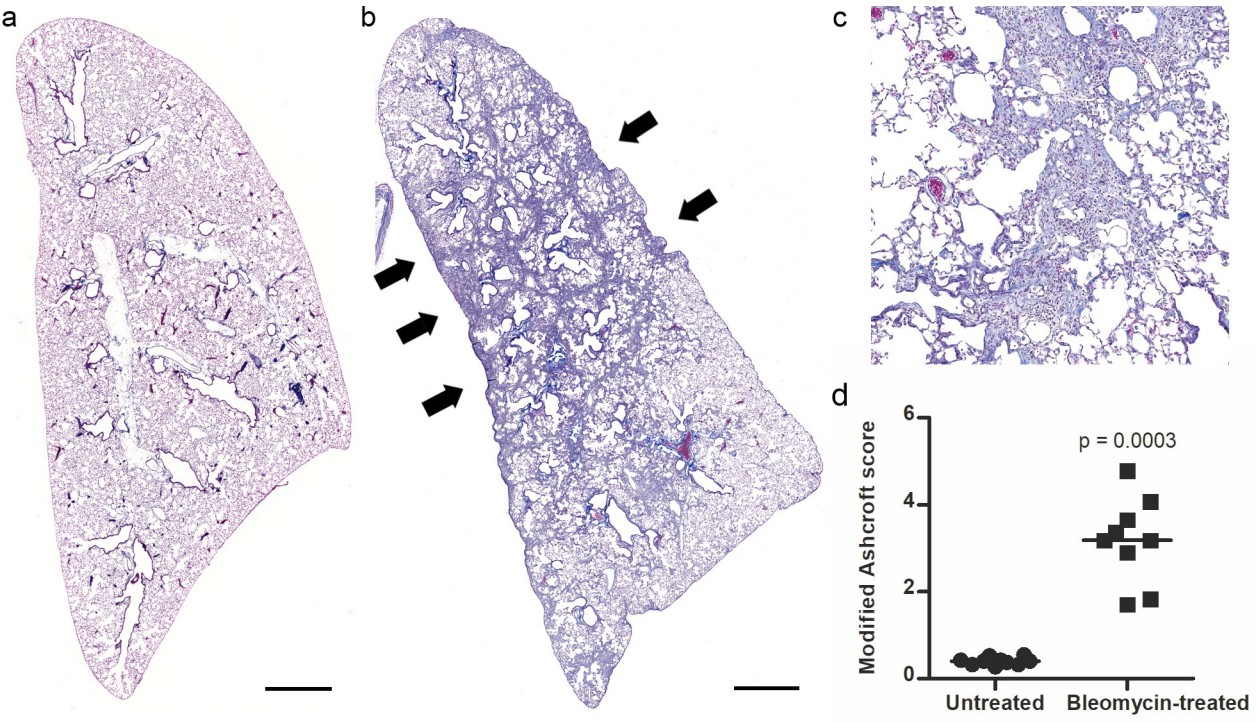

**Fig 1. Lung fibrosis at 4 weeks after bleomycin administration.** a) Masson's trichrome staining of an untreated left lung and b) of a bleomycin-treated left lung. Fibrotic changes are indicated by arrows. Scale bars 1 mm. c) Example of a 0.9 mm wide close-up image of a bleomycin-treated lung produced by the random sampling procedure, with a modified Ashcroft score of 5. A nonlinear tone curve adjustment was applied to all images to correct white balance. d) Modified Ashcroft scores for the experimental groups, untreated (round symbols) and bleomycin-treated (square symbols). Lines at means. n = 9–10. Mann-Whitney U test.

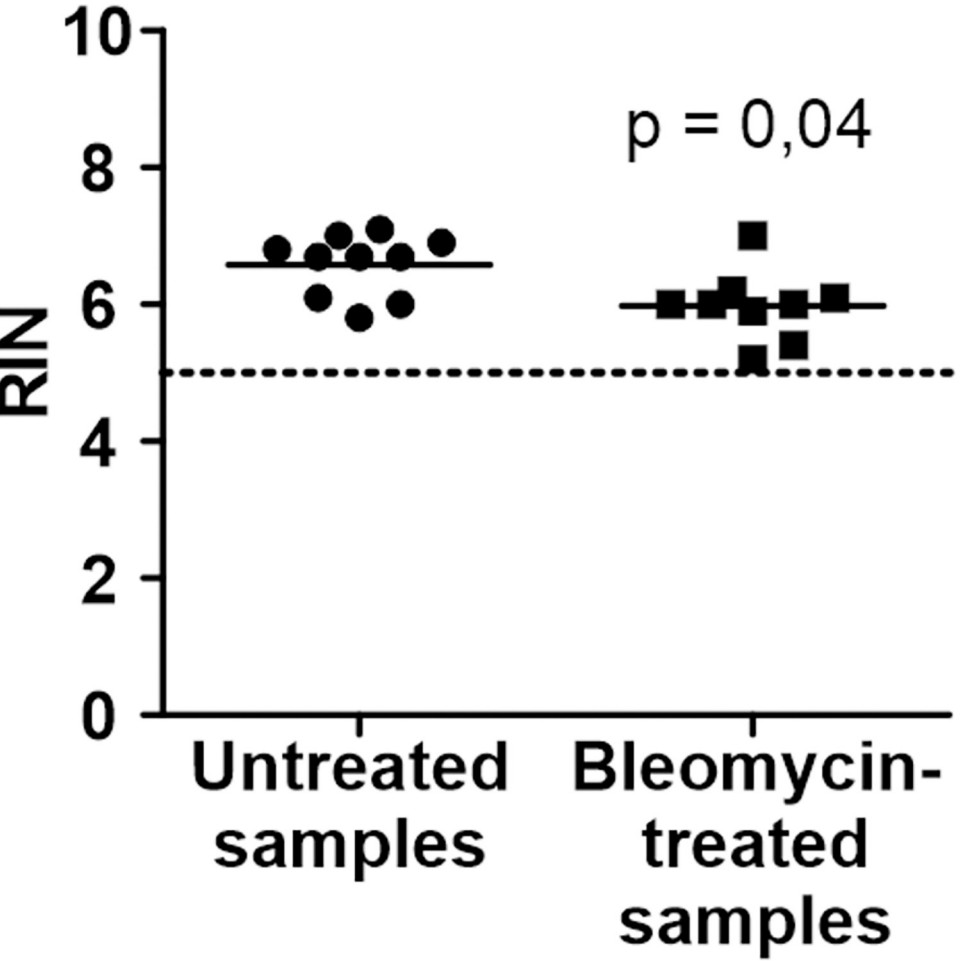

**Fig 2. RNA quality.** RIN values for the untreated (round points) and bleomycin-treated samples (square points). Lines at means. n = 9–10. Mann-Whitney U test. Dashed line at RIN = 5.

### RNA integrity

Given that RNA integrity (RIN) values above 5 are considered sufficient for RT-qPCR analysis [30], the average RIN values recorded here were 6.6 (SD 0.45) for untreated and 6 (SD 0.51) for bleomycin-treated samples (Fig 2). The values were on average 10%, or 0.6 units, lower in the bleomycin-treated group (Fig 2), possibly reflecting different conditions in the fibrotic tissue, since the sample collection and RNA extraction procedures were the same for all the samples. Nevertheless, given the similarity of the values, the samples should be readily comparable.

### Expression characteristics of the candidate genes

The primer efficiency was between 85% and 110% for all the primer pairs, which is considered sufficient for RT-qPCR [30]. This efficiency was taken into account when correcting the Cq values and when calculating the relative quantities. All the primer pairs amplified products that had efficiency-corrected Cq values between 13 and 30 (Fig 3), and each primer pair showed uniform single-peak melting curves for all the samples. No template control (NTC) Cq values were over 38 for all the primers except for *Rn18s*, for which they were above 30, and for *Sdha*, for which they were above 33, all indicating sufficiently low background signal levels. No

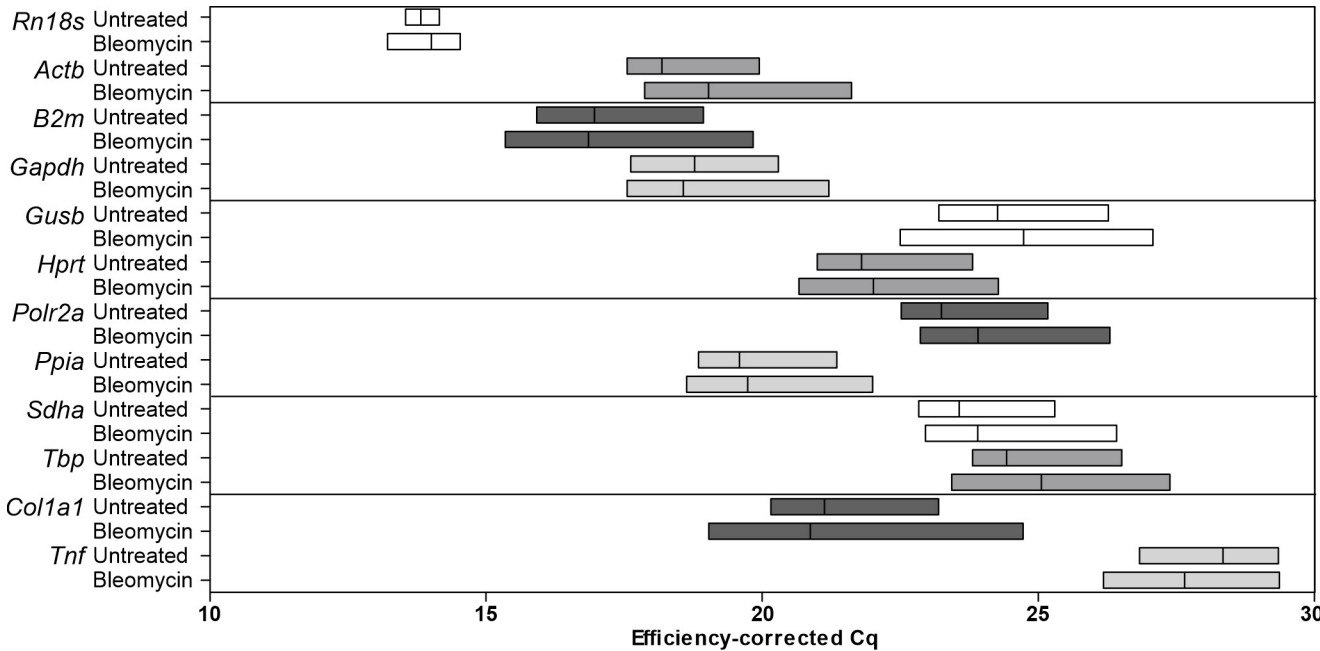

**Fig 3. Efficiency-corrected Cq values for the genes investigated in untreated and bleomycin-treated mice.** Untreated: untreated samples. Bleomycin: bleomycin-treated samples. Each box has lines for minimum, maximum and mean values. X-axis truncated; 40 cycles were run. n = 9–10.

reverse transcription (NRT) controls for *Gapdh* had a Cq value of 22.13 while the highest Cq for a sample was 21.21 and *Ppia* had a Cq of 23.90 with the highest Cq for a sample being 22.0. NRT controls for all other genes had Cq values at least 7 cycles higher than the sample with the highest Cq, indicating that the primers did not significantly amplify genomic DNA. The NRT values for *Gapdh* and *Ppia* indicate that up to 35% and 22%, respectively, of the expression detected with RT-qPCR could originate from pseudogenes or other unspecific amplification and cause variation in the results. Neither the primers for *Gapdh* nor those for *Col1a1* were exon junction-spanning, but as the NRT control for *Col1a1* did not exhibit unspecific amplification from genomic DNA, the unspecific amplification in the NRT control for *Gapdh* is most likely to have originated from pseudogenes or similar sequences.

### Reference gene stability

Normfinder and geNorm for Bio-Rad CFX Maestro algorithms were used to rank the reference genes in terms of their stability, and averages were calculated from these ranks to obtain the overall ranking (Fig 4). *Sdha* performed best in both analyses, and *Ppia*, *Polr2a* and *Hprt* were the three next best genes, although the potential pseudogene problem of *Ppia* makes it a suboptimal choice. The most commonly used reference gene, *Gapdh*, showed the poorest performance according to both programs, and *Actb* and *Rn18s* performed less well than most other candidates. Thus we suggest that *Sdha*, *Polr2a* and *Hprt* would form the best combination of reference genes for assessing bleomycin-induced pulmonary fibrosis at the 4-week time point.

### Validation of candidate reference genes

In order to validate the findings, the expression levels of *Col1a1* and *Tnf*, which are expected to be increased under fibrotic conditions, were compared between the experimental groups using

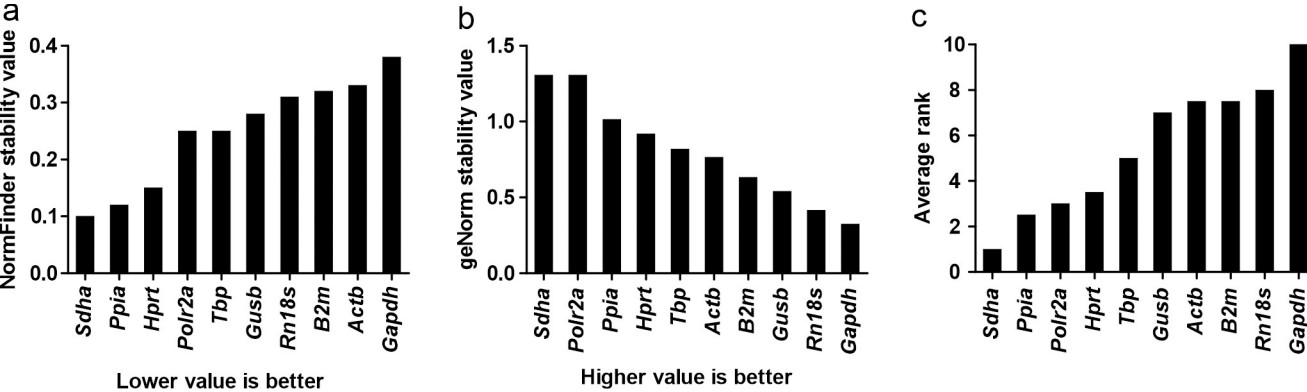

**Fig 4. Comparison of the stability values of the candidate reference genes.** Values from a) NormFinder and b) geNorm, and c) average of ranks based on the two programs.

*Sdha*, *Polr2a* and *Hprt* as the reference genes (Fig 5). The expression of *Col1a1* appears to be slightly upregulated in the bleomycin-treated samples and *Tnf* clearly so, as expected. When only *Gapdh* was used as a reference gene, however, neither *Col1a1* nor *Tnf* showed any significant differences between the groups.

## Discussion

We present here a strategy for determining the optimal reference genes for RT-qPCR normalization in the widely used bleomycin model of pulmonary fibrosis at 4 weeks post-treatment, i.e. the late fibrotic stage. Accurate normalization is the key to obtaining reliable results, and selecting stably expressed reference genes for normalization can help avoid excessive variation in the results. As established in many earlier studies, a single reference gene is rarely enough for reliable results, so that at least three genes should be used after determining the invariability of their expression [9, 31, 32].

The candidate genes we selected include some of the most popular reference genes used in RT-qPCR analyses, genes having various functions in the cell's basic activities that are integral to its survival, and therefore genes that may be expected to be expressed steadily under different conditions. We picked primer sequences that were found to work well in previously

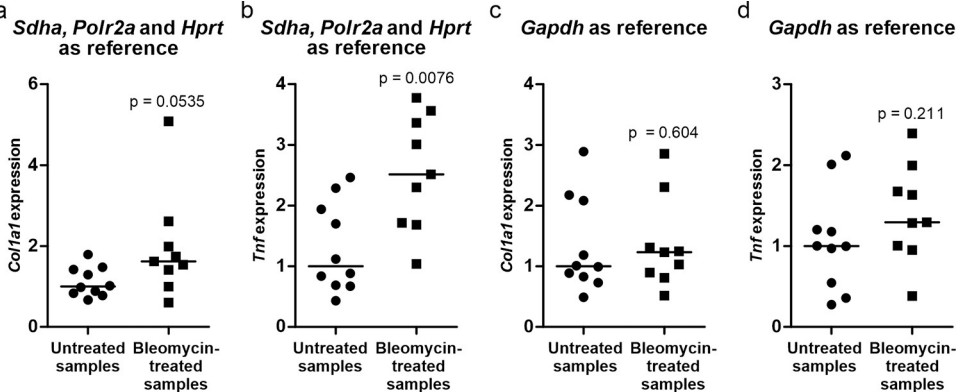

**Fig 5. Validation of selected normalization strategies.** Changes in the expression of *Col1a1* (a, c) and *Tnf* (b, d) in bleomycin-treated mice (square points) relative to untreated mice (round points) when *Sdha*, *Polr2a* and *Hprt* are used as reference genes (a, b) and *Gapdh* as reference (c, d). Lines at medians. n = 9–10. Mann-Whitney U test.

published studies and were careful to select primers that were specific to the gene of interest, especially with *Actb*. This is because some of the primers used previously, and also some commercially available primers, were predicted to be unspecific and also to amplify *Lrrc58*, with which *Actb* shares 91% of its sequence. The primers for *Gapdh* and *Ppia*, however, were found to be nonspecific, as seen from their high NRT values, and hence were deemed unreliable and nonoptimal for acting as reference genes. The fact that the NTC for both genes failed to show significant amplification argues against primer dimerization, and thus the amplification may have been derived from pseudogenes. The pseudogene problem attached to *Gapdh* has been demonstrated earlier [5], but its continued ubiquitous use as a reference gene warranted its inclusion in this study. The results further demonstrate that the use of *Gapdh* as the sole reference gene is not appropriate without determining the invariability of its expression by comparing it with a few alternatives, although we would rather recommend reference genes that are free of pseudogene issues. As shown in Fig 5, inappropriate normalization could lead to false results. *Actb* and *Rn18s*, the other most commonly used reference genes [4], showed poor performance compared with the remaining candidates studied here. The main issue with *Rn18s* is that the product is rRNA instead of mRNA and thus its expression is governed by different factors from those lying behind the genes of interest [32].

In this study we used two different algorithms to determine the stability of 10 candidate genes in the late fibrotic phase of bleomycin-induced pulmonary fibrosis in mice. Although the two algorithms represent different approaches to calculating reference gene stability, the best-performing genes are the same in both programs, a fact that highlights their stability. A previous study focusing on the initial inflammatory phase in the bleomycin-induced pulmonary fibrosis model found *Hprt*, *Ppia* and *Sdha* to be the most stable reference genes [13], and our results confirmed the very same genes, with the addition of *Polr2a*, as acting reliably as reference genes in RT-qPCR at a later stage in pulmonary fibrosis. Taken together, we propose that *Hprt*, *Polr2a* and *Sdha* form a reliable set of reference genes for RT-qPCR in the bleomycin model of pulmonary fibrosis.

## Supporting information

**S1 Table. Data underlying the findings described.** Cq values and plots of standard curves used for determination of primer efficiency, raw Cq values for all samples, efficiency-corrected Cq values and relative quantities, datafile used for NormFinder analysis, and individual values behind the graphs and results.
(XLSX)

## Acknowledgments

The authors thank Jaana Peters and Päivi Tuomaala for technical assistance and Oulu Laboratory Animal Center (OULAC) for care of the animals. The work was supported by Biocenter Oulu, Sequencing Center, and the Transgenic and Tissue Phenotyping Core Facility endowed by the University of Oulu, Finland, and Biocenter Finland.

## Author Contributions

**Conceptualization:** Oula Norman, Jarkko Koivunen, Joni M. Mäki, Taina Pihlajaniemi, Anne Heikkinen.

**Data curation:** Oula Norman.

**Formal analysis:** Oula Norman.

**Funding acquisition:** Oula Norman, Taina Pihlajaniemi.

**Investigation:** Oula Norman, Jarkko Koivunen, Anne Heikkinen.

**Methodology:** Oula Norman, Jarkko Koivunen, Joni M. Mäki.

**Project administration:** Anne Heikkinen.

**Resources:** Taina Pihlajaniemi.

**Supervision:** Taina Pihlajaniemi, Anne Heikkinen.

**Validation:** Oula Norman.

**Visualization:** Oula Norman.

**Writing – original draft:** Oula Norman, Anne Heikkinen.

**Writing – review & editing:** Oula Norman, Jarkko Koivunen, Joni M. Mäki, Taina Pihlajaniemi, Anne Heikkinen.

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
