## [Decision Letter · Decision Letter 0]

6 Sep 2022

PONE-D-22-22612Identification of suitable reference genes for normalization of reverse transcription quantitative real-time PCR in the fibrotic phase of the bleomycin mouse model of pulmonary fibrosisPLOS ONE

Dear Dr. Heikkinen,

Thank you for submitting your manuscript to PLOS ONE. After careful consideration, we feel that it has merit but does not fully meet PLOS ONE’s publication criteria as it currently stands. Therefore, we invite you to submit a revised version of the manuscript that addresses the points raised during the review process.

We look forward to receiving your revised manuscript.

Kind regards,

Kyung-Wan Baek, Ph.D.

Academic Editor

PLOS ONE

Additional Editor Comments :

Your manuscript will be considered suitable for publication in PLos ONE if it undergoes appropriate revisions. Please revise the manuscript to reflect the reviewers' comments.

Reviewers' comments:

Reviewer's Responses to Questions

**Comments to the Author**

1. Is the manuscript technically sound, and do the data support the conclusions?

Reviewer #1: Yes

Reviewer #2: Yes

2. Has the statistical analysis been performed appropriately and rigorously? 

Reviewer #1: Yes

Reviewer #2: Yes

3. Have the authors made all data underlying the findings in their manuscript fully available?

Reviewer #1: Yes

Reviewer #2: Yes

4. Is the manuscript presented in an intelligible fashion and written in standard English?

Reviewer #1: Yes

Reviewer #2: Yes

5. Review Comments to the Author

Reviewer #1: This paper presents proper reference genes for the RT-qPCR in the pulmonary fibrosis mouse model after treatment of the Bleomycin. In the fibrosis mouse model, the reference gene selection is an important and appropriate study for verifying gene expression. If the authors solve the minor issues below, this paper will be published to the PLoS One.

Manuscript Overall: Gene Symbol should be used as an italic, and all capital letters.

Line 2: Please provide abbreviation of reverse transcription quantitative real-time PCR as (RT-qPCR) in title.

Line 17: two separate algorithms → Please present specific names.

Line 85-86: “ab libitum” please provide as italic.

Line 130 Table 1, 5th and 7th columns: please provide direction of primer sequences (5’-3’).

Figure 3 and 5: There may be confusion in sample names. “Untreated” → “Untreated samples”, “Bleomycin” → “Bleomycin-treated samples”. If you lack space, you can use abbreviations. "UC" and "BTS". The difference between the groups according to the shape of each point (round point, square point) should be described in each figure legend.

Reviewer #2: The ms. by Norman et al. presents data regarding a number of genes that could serve as reference genes in the bleomycin mouse model. These are proposed as stable denominators for gene expression assays during development of fibrosis. The results are an incremental advance of interest to a limited number of investigators. The methods appear technically sound and the data might prove to have some utilitarian value.

6. PLOS authors have the option to publish the peer review history of their article (what does this mean?). If published, this will include your full peer review and any attached files.

Reviewer #1: No

Reviewer #2: No

---

## [Author Response · Author response to Decision Letter 0]

29 Sep 2022

We highly appreciate the opportunity to improve our manuscript according to the valuable comments highlighted by the Reviewers. We have done our best to fully respond to the concerns raised by the reviewers.

Reviewer #1: This paper presents proper reference genes for the RT-qPCR in the pulmonary fibrosis mouse model after treatment of the Bleomycin. In the fibrosis mouse model, the reference gene selection is an important and appropriate study for verifying gene expression. If the authors solve the minor issues below, this paper will be published to the PLoS One.

1) Manuscript Overall: Gene Symbol should be used as an italic, and all capital letters.

RESPONSE: In our manuscript we have followed the gene nomenclature used by the National Center for Biotechnology Information (NCBI), see e.g., the gene for mus musculus collagen type I α1 chain (https://www.ncbi.nlm.nih.gov/gene/12842). According to this, mouse genes are written in a way that they start with an uppercase letter and all other letters are given by lowercase letters. This style is used to differentiate rodent genes from human genes, which are written in capital letters. Official gene symbols for mouse genes are provided by the Mouse Genome Informatics at the Jackson laboratory (http://www.informatics.jax.org/mgihome/nomen/gene.shtml#gaas), as follows: 

“Guidelines for Nomenclature of Genes, Genetic Markers, Alleles, and Mutations in Mouse and Rat

2.3 Gene Symbols and Names

2.3.1 Gene Symbols

Genes are given short symbols as convenient abbreviations for speaking and writing about the genes.

A gene symbol should:

• be unique within the species and should not match a symbol in another species that is not a homolog.

• be short, normally 3-5 characters, and not more than 10 characters

• use only Roman letters and Arabic numbers

• begin with an uppercase letter (not a number), followed by all lowercase letters / numbers (see exception below)

• not include tissue specificity or molecular weight designations

• include punctuation only in specific special cases (see below)

• ideally have the same initial letter as the initial letter of its gene name to aid in indexing. However, letter order in a gene symbol need not follow word order in the name.

• When a definitive human ortholog exists, gene symbols should agree with human gene symbols when practical.

• Never prepend 'm' (for a mouse gene symbol) or 'r' (for a rat gene symbol) to indicate species.”

Based on this convention, we would like to stay with the original gene symbol nomenclature, if acceptable by the reviewer. All gene names were originally in italic, but we noticed that when submitting, the system converted italic font to not italic in the Abstract and we were not able to correct this. All gene names are (and were originally) in italic in the Word version of the manuscript.

2) Line 2: Please provide abbreviation of reverse transcription quantitative real-time PCR as (RT-qPCR) in title.

RESPONSE: The abbreviation (RT-qPCR) has been added in the title; page 1, line 5.

3) Line 17: two separate algorithms → Please present specific names.

RESPONSE: Names of two separate algorithms are now present in the Abstract, page 2, line 31.

4) Line 85-86: “ab libitum” please provide as italic.

RESPONSE: “ab libitum” is provided as italic, page 5, lines 101-102.

5) Line 130 Table 1, 5th and 7th columns: please provide direction of primer sequences (5’-3’).

RESPONSE: Directions of primer sequences (5’-3’) has been provided on 5th and 7th columns, page 7, Table 1, row 1.

6) Figure 3 and 5: There may be confusion in sample names. “Untreated” → “Untreated samples”, “Bleomycin” → “Bleomycin-treated samples”. If you lack space, you can use abbreviations. "UC" and "BTS". The difference between the groups according to the shape of each point (round point, square point) should be described in each figure legend.

RESPONSE: The requested change “Untreated” → “Untreated samples”, “Bleomycin” → “Bleomycin-treated samples” has been made in the Fig 2 and Fig 5. Fig 3 itself has not been changed but the following explanation; “Untreated: untreated samples. Bleomycin: bleomycin-treated samples” has been added in the Fig 3 legend, page 10, line 219. 

The difference between the groups according to the shape of each point; untreated (round symbols) and bleomycin-treated (square symbols) is now described in the Fig 1 (page 9, lines 188-189), 2 (page 9, lines 199-200) and 5 (page 11, lines 242-243) legends.

Reviewer #2: The ms. by Norman et al. presents data regarding a number of genes that could serve as reference genes in the bleomycin mouse model. These are proposed as stable denominators for gene expression assays during development of fibrosis. The results are an incremental advance of interest to a limited number of investigators. The methods appear technically sound and the data might prove to have some utilitarian value.

---

## [Editor Report · Decision Letter 1]

2 Oct 2022

Identification of suitable reference genes for normalization of reverse transcription quantitative real-time PCR (RT-qPCR) in the fibrotic phase of the bleomycin mouse model of pulmonary fibrosis

PONE-D-22-22612R1

Dear Dr. Heikkinen,

We’re pleased to inform you that your manuscript has been judged scientifically suitable for publication and will be formally accepted for publication once it meets all outstanding technical requirements.

Kind regards,

Kyung-Wan Baek, Ph.D.

Academic Editor

PLOS ONE

Additional Editor Comments (optional):

It appears that the manuscript has been improved to a level suitable for publication in PLoS ONE. I do not believe that further review is necessary.
---

## [Editor Report · Acceptance letter]

7 Oct 2022

PONE-D-22-22612R1 

Identification of suitable reference genes for normalization of reverse transcription quantitative real-time PCR (RT-qPCR) in the fibrotic phase of the bleomycin mouse model of pulmonary fibrosis 

Dear Dr. Heikkinen:

I'm pleased to inform you that your manuscript has been deemed suitable for publication in PLOS ONE. Congratulations! Your manuscript is now with our production department. 

Kind regards, 

on behalf of

Dr. Kyung-Wan Baek 

Academic Editor

PLOS ONE